Easier said than done: unexpected hurdles to preparing ∼1,000 cranial CT scans for data collection from an online digital repository

Modesto-Mata Mario paleomariomm@gmail.com 1 2
Thiebaut Arthur 1
Krueger Kristin L. 3
Maga A. Murat 4
Joganic Jessica L. 5
Ryan Timothy M. 6
Richtsmeier Joan T. 6
Cheverud James M. 7
Hlusko Leslea J. leslea.hlusko@cenieh.es hlusko@berkeley.edu 1
1 Centro Nacional de Investigación sobre la Evolución Humana , Burgos , Castilla y León , Spain
2 Universidad Internacional de La Rioja , Logroño , La Rioja , Spain
3 Department of Anthropology, Loyola University of Chicago , Chicago , IL , United States of America
4 Department of Pediatrics, University of Washington School of Medicine , Seattle , WA , United States of America
5 National Park Service , Ashland , WI , United States of America
6 Department of Anthropology, Pennsylvania State University , University Park , PA , United States of America
7 Department of Biology, Loyola University of Chicago , Chicago , IL , United States of America
De Baets Kenneth
Electronic publication date: 2025 Oct 20
Publication date: 2025
Volume: 13
Electronic Location ID: e20172
Received 2025 May 15; Accepted 2025 Sep 11
Copyright: ©2025 Modesto-Mata et al.
Copyright year: 2025
Copyright holder: Modesto-Mata et al.
License: This is an open access article distributed under the terms of the Creative Commons Attribution License, which permits unrestricted use, distribution, reproduction and adaptation in any medium and for any purpose provided that it is properly attributed. For attribution, the original author(s), title, publication source (PeerJ) and either DOI or URL of the article must be cited.
License URL: https://creativecommons.org/licenses/by/4.0/

Keywords: Data-collection, Slice spacing, 3d slicer, CT scans, DICOM, Public data repositories, Digital imaging validation, Metadata accuracy, Quantitative genetics, Craniodental variation

Funding: European Research Council within the European Union’s Horizon Europe ERC-2021-ADG, Tied2Teeth, project number 101054659 MCIN/ AEI/10.13039/501100011033/ FEDER, UE Office of Research Infrastructure Programs, National Institutes of Health The National Science Foundation grant 2007-10 This research was funded by the European Research Council within the European Union’s Horizon Europe (ERC-2021-ADG, Tied2Teeth, project number 101054659). Research at the CENIEH is supported by funding from the Ministerio de Ciencia e Innovación Project PID2021-122355NB-C33 financed by MCIN/ AEI/10.13039/501100011033/ FEDER, UE. This investigation used resources that were supported by the Southwest National Primate Research Center grant P51 OD011133 from the Office of Research Infrastructure Programs, National Institutes of Health, and also by the National Science Foundation grant 2007-10 (Developmental Genetic basis of primate craniofacial variation, and human origins BCS-0725227). There was no additional external funding received for this study. The funders had no role in study design, data collection and analysis, decision to publish, or preparation of the manuscript.

==============================
Background

As science becomes more open and accessible, researchers are increasingly encouraged—and sometimes required—to share their digital data on public repositories. While this promotes transparency and reusability, it can also introduce challenges. We highlight one such challenge by detailing our experience processing computerized tomography (CT) scans of 985 baboon skulls downloaded from MorphoSource, part of a quantitative genetic study of craniodental variation in the pedigreed baboon colony from the Southwest National Primate Research Center. When importing DICOM files into 3D Slicer, 182 of the 985 scans (18.5%) generated an “inconsistent slice spacing” error. When prompted, 3D Slicer “corrected” this by regularizing the slice spacing. However, this led to a mismatch between the slice spacing reported on MorphoSource and the spacing adjusted by 3D Slicer.

Methods

To determine which slice spacing was accurate, we compared Prosthion-Basion (PR-BA) distances measured directly from physical skulls (using calipers and a Microscribe) with those derived from the CT models. We ran paired t-tests to determine whether there were differences between them. Our comparison sample included five randomly selected skulls from the error group and fifteen ramdon skulls from the error-free group (which exhibited various slice spacings when scanned).

Results

For scans without the slice spacing error, there was strong agreement between physical and digital measurements, indicating metadata accuracy. For error-generating scans, measurements based on 3D Slicer’s corrected spacing and Amira-Avizo both aligned well with the physical data. In contrast, manually overriding the spacing to match the MorphoSource metadata led to overestimations of the PR-BA distance.

Conclusion

Although the discrepancy was straightforward to describe, resolving it required over 250 person-hours across 8 months. Accessing physical specimens, conducting repeated measurements, and cross-validating with multiple tools made the process labor-intensive. Nonetheless, this effort avoided a 3–5% measurement bias in nearly 20% of our sample and allowed inclusion of these scans in downstream semi-automated data collection. We urge researchers to thoroughly understand the digital datasets they work with and resist the temptation to ignore apparent errors during import. We also recommend that funding bodies provide support for the extensive time needed to validate and process digital imagery, both for data generators and users. Finally, we highlight the need for public repositories to implement stronger quality control. If a data import check similar to 3D Slicer’s had been applied during data submission, the inconsistency between manually entered metadata and embedded DICOM information might have been caught and corrected at the time of upload.

Introduction

Development of digital specimen repositories has significantly advanced the goal of data-sharing and democratization of the anatomical sciences by making immense numbers of scanned biological specimens available to any researcher with access to the internet (Weber et al., 2001; Bradtmöller et al., 2010; Copes et al., 2016; Boyer et al., 2016; Lebrun & Orliac, 2016; Davies et al., 2017; Blackburn et al., 2024). For example, the data aggregator iDigBio (Nelson, 2014) enables the research community to have access to millions of biological specimens in digital format (Nelson & Paul, 2019). Taxa available for study are incredibly diverse and include fishes (Singer, Love & Page, 2018), bats (Shi, Westeen & Rabosky, 2018), non-human primates (Copes et al., 2016; Barger et al., 2021), reptiles (Uetz et al., 2024) and human medical imaging (Vannier, Staab & Clarke, 2002; Clark et al., 2013; Edgar et al., 2020; Lebre, Silva & Costa, 2020). The research potential of these 3D digital repositories is immeasurable, although the use of images from human subjects comes with complicating ethical factors (Schug et al., 2020; Spake, Nicholas & Cardoso, 2020).

Traditional measurement methods provide a striking contrast with the modalities available for the analysis of these virtual models. In the past, physical specimens were measured directly or from high-resolution analog replicas; cranial and endocranial distances and brain sizes were obtained via physical instruments (Pickering, 1930; Wagner, 1935) or radiographs (Hansman, 1966), and mesiodistal and buccolingual tooth diameters were measured with calipers (Garrod et al., 1928; Garn, Lewis & Kerewsky, 1966; Boklage, 1987). It was not until the final decades of the 20th century that laser scanning in paleoanthropology made 3D models available for virtual measurements (Zollikofer, Ponce De León & Martin, 1998), and tomographic and microtomographic techniques (CT and mCT) transformed fossil analysis and data distribution (Sutton, 2008; Wu & Schepartz, 2009). Today, virtual reconstructions of fossils allow for the acquisition of previously inaccessible measurements, for example, enabling researchers to calculate enamel and dentine surfaces and volumes in fossilized dentitions (Martínez de Pinillos et al., 2017; García-Campos et al., 2019; Martín-Francés et al., 2020), analyze the internal structures of cranial bones such as diploic channels (Lázaro et al., 2020), and explore the auditory capacities of ancient taxa through cochlear studies (Conde-Valverde et al., 2019), just to name a few. Additionally, virtual models support the broad application of new statistical methods, including geometric morphometrics (Bruner, 2004; Bastir et al., 2017; Palancar et al., 2021) and artificial intelligence techniques (Yu et al., 2022; Yu et al., 2024).

Resources resources are available for guidance on how to prepare image files prior to upload and advice on how to follow the F.A.I.R. principles: Findable, Accessible, Interoperable, and Reusable (Wilkinson et al., 2016; Davies et al., 2017; Jacobsen et al., 2020). Additionally, there are resources that guide investigators on how to utilize CT scans downloaded from these repositories (Buser et al., 2020), urge caution with respect to the sources of 3D measurement error (Shearer et al., 2017), and provide warnings about compiling 3D data from other researchers (Robinson & Terhune, 2017). However, in our review of the literature, we were unable to find scientific publications specifically aimed at providing perspective on the time investment needed to prepare data prior to making them publicly available, or shedding light on the challenges and time investment that researchers may need to prepare 3D scans from open-source data repositories.

Here, we share our experience preparing 985 CT scans of baboon crania downloaded from MorphoSource, a public image repository for data collection (Boyer et al., 2016). These scans derive from one population of baboons that is part of a pedigreed breeding colony at the Southwest National Primate Research Center (SNPRC), a colony that has been used in quantitative genetic analyses (Rogers et al., 2000; Cox et al., 2006) for phenotypes that include dental variation (Hlusko et al., 2004; Hlusko et al., 2016), craniofacial variation (Sherwood et al., 2008; Willmore et al., 2009; Roseman et al., 2010; Joganic et al., 2018), cardiovascular disease (Mahaney et al., 2018), bone density (Havill et al., 2010), life span (Martin et al., 2002), and even dimensions of personality (Johnson et al., 2015). Additionaly, these baboons have been widely used to answer research questions about genomics (Spradling et al., 2013; Robinson et al., 2019; Kos et al., 2021), pathology (Szabó & Salinas, 2021), microbiology (Tsukayama et al., 2018), and brain architecture (Atkinson et al., 2015).

Given the large sample size needed for quantitative genetic analyses, we planned to automate parts of the phenotype data collection process to save time (Boukamcha et al., 2017; Bannister et al., 2020; Kang et al., 2021; Nguyen et al., 2022). Recently, a new pipeline has been developed using automatic landmarking via multiple templates (MALPACA) (Zhang et al., 2022). The process is deployed as a module in the Slicermorph extension (Rolfe et al., 2021) that runs in 3D Slicer (Fedorov et al., 2012) and has been used successfully to analyze zebrafish models (Diamond et al., 2023). In preparation of the 985 SNPRC CT scans for use with MALPACA, we first imported the DICOM files to 3D Slicer. It was during this step that we ran into a hurdle that required an unexpected and extended investment of time.

We present our journey in this article for four primary reasons. First, we want to provide the solution, which required access to physical specimens, so that future users of these CT scans will know how to modify the files accordingly. Second, our experience can serve as a cautionary tale for others when they are anticipating the amount of time that may be needed to prepare CT scans for data collection. Third, we provide a warning for future researchers to thoroughly understand their digital datasets and question every potential inconsistency or error. And fourth, we hope that this situation will provide motivation for colleagues to ask funding agencies for adequate support for preparing and uploading their CT scans to a digital repository, as part of following the best practices for publishing verified 3D digital data (Davies et al., 2017).

Materials & Methods

The baboon colony and skull collection

The baboon (genus Papio) skeletal sample came from a colony maintained by the Southwest National Primate Research Center (SNPRC), located at the Texas Biomedical Research Institute in San Antonio (Texas, USA). The founders of this colony were wild baboons caught in southwestern Kenya, in a hybrid area between two subspecies: olive baboons (P. hamadryas anubis) and yellow baboons (P. h. cynocephalus) (Maples & McKern, 1967). The majority of founders were from the former subspecies. More than 2,400 individuals out of the roughly 21,000 who have resided within the SNPRC colony form a single, complex pedigree for which their kinship relations are well documented (Rogers et al., 2000; Hlusko, Weiss & Mahaney, 2002; Joganic et al., 2018). Traditionally, all members of Papio were interpreted as one species, P. hamadryas, and the different geographic variants were considered different subspecies (Jolly, 2003). This is the approach followed by the SNPRC. More recent taxonomic practice is to divide Papio into six different species with significant hybridization between them (Boissinot et al., 2014). We adopt here the naming convention of the SNPRC, as in this paper the taxonomy of this genus is not the aim.

After death, each baboon was necropsied by SNPRC veterinarians, skeletonized via maceration or with dermestid beetles, and their skulls archived at Washington University in St. Louis (WUSTL) under the curation of J.M.C. and J.L.J. While residing at the SNPRC, each animal was assigned a four-, five-, or six-digit alphanumeric identification number. During the transition from the SNPRC to the WUSTL skeletal collection, each skull was given a new specimen number beginning with “W” and running from W001 to W985. These skulls were later transferred to Loyola University in Chicago but are now in the process of being shifted to the University of Illinois Urbana-Champaign (to be maintained by Charles Roseman).

The cranial CT scans

Nine hundred and eighty-five skulls were imaged using a Siemens Biograph 40 TruePoint Tomograph at the Center for Clinical Imaging Research at Washington University School of Medicine. The resulting CT images were uploaded to MorphoSource between March 2018 and August 2019 (https://www.morphosource.org/projects/00000C475).

The term “slice thickness” reports the amount of anatomical information contained within a single CT image or slice. This contrasts with “slice spacing”, the distance between the center of two consecutive DICOM slices. Slice spacing and thickness can be the same or different, depending on the acquisition parameters. However, the geometry of the volume (3D data) is determined by the slice spacing parameter. For these baboon skull CT images, these parameters are the same, and because the term “slice spacing” is used in 3D Slicer, this is the term that we employ here.

According to the CT scan metadata provided on MorphoSource, individuals W001 to W487 were scanned at a slice spacing of 0.75 mm, and individuals W488 to W985 were scanned with a slice spacing of 0.60 mm. Willmore et al. (2009), Roseman et al. (2010), Joganic et al. (2018), and Joganic & Heuzé (2019) analyzed data collected from a subset of the skulls scanned with the 0.75 and 0.60 mm slice spacing, and their description of the scanner and scan settings agree with the information provided on MorphoSource. Atkinson et al. (2015) analyzed data from the full data set of 985 individuals but reported that the slice spacing was either 0.6 or 0.7 mm for all specimens (Atkinson et al., 2015), and that they used a General Electric 3D CT scanner. However, the metadata described by MorphoSource, and reported by Willmore et al. (2009) and Roseman et al. (2010) indicates 0.75 mm slice spacing and that a Siemens scanner was used. We interpret the differing information about slice spacing and scanner specs in Atkinson et al. (2015) to be typographic errors.

The CT images of each skull were oriented in the frontal plane, beginning at the anterior part of the face (including the incisors) and progressing posteriorly to the occipital region and sagittal crest, if present. As a result, any measurement along the anterior-posterior axis is mostly influenced by the slice spacing, whereas width or height distances are less sensitive to this parameter.

Reading the DICOM files: identification of the slice spacing problem

We used the 3D Slicer software (Fedorov et al., 2012) for scan processing. This is a free, open-source program for visualization, processing, segmentation, registration, and analysis of 3D images and meshes.

Initially, all skull CT scans were expected to be imported into 3D Slicer following the same sequence of steps with the DICOM Import module, regardless of specific slice spacing used during the initial image acquisition process at WUSTL. This involves navigating to the directory where the skull DICOM files are located; selecting the patient (individual), study, and series; and clicking on “Load”, resulting in the generation of a single volume that includes the entire skull, as shown for skull W201 (Fig. S1). We were able to successfully follow this process for most of the specimens: all skulls scanned with a 0.75 mm slice spacing (W001–W487) and most of those scanned with a 0.6 mm slice spacing (W670–W985). However, an import warning with an error was generated for 182 of the scans with 0.6 mm slice spacing.

When scans for W488 to W669 were loaded into 3D Slicer, each skull generated ∼50 volumes instead of one. The number of volumes is different from one specimen to another, depending on the size of the skull. For instance, 51 volumes were created for the specimen W584 (Fig. S2). When we visualized one of these 51 volumes (e.g., “2: InnerEarSeq 0.6 U75u–acquisitionNumber 39”), we realized that only part of the skull was displayed, and this volume was comprised of six slices (Fig. S3).

In order to load the entire skull for these multivolume specimens, we navigated to the Advanced option in the DICOM module and selected Examine. On that screen, we unchecked all 51 volumes of W584 and then re-checked only the last one, named in our example “2: InnerEarSeq 0.6 U75u” (Fig. S4).

After clicking on “Load” a warning message appeared indicating that “0.6 spacing was expected, 0.5 spacing was found between files [...]” (Fig. S5). Eventually, the entire skull for W584 was loaded, but with a slice spacing of 0.58360656 mm, rather than the expected 0.60 mm (Fig. S6). The resultant spacing is different depending on the skull, as this value is automatically calculated by 3D Slicer based on the number of volumes per skull and slices within each volume. In these cases, within a volume there is a constant spacing of 0.60 mm, but between volumes there is a spacing of 0.50 mm. In all cases, the first two digits of the slice spacing are a constant (0.58) and spacing differences become apparent in the thousandths position and beyond.

The effect of the slice spacing mismatch on data collection

To demonstrate the effect of an erroneous slice spacing assignment of 0.58 mm instead of 0.60 mm on anatomical data collection and eventual measurement, we continue to use skull W584 as an example. As described above, we loaded the W584 DICOM files in 3D Slicer, which automatically assigned a slice spacing of 0.58360656 mm.

Using a slice spacing of 0.58360656 mm the linear distance between a small fracture on the incisal edge of the upper left central incisor and the posterior-most point of the skull in sagittal position (Fig. S7A) was estimated to be 173.3 mm. We then manually changed the slice spacing to 0.60 mm to agree with the slice thickness indicated in the original DICOM header file of this specimen (Fig. S7B). Using these data, we collected the 3D coordinates of the same two landmarks and the linear distance between them was estimated to be 178.2 mm indicating a 4.9 mm difference between the same linear distance estimated on the image data that differed only in the assigned slice spacing (Fig. S7C). In other words, the linear distance estimated on the 0.60 mm model for specimen W584 is 2.83% larger than that estimated from the 0.58 mm model. As quantitative genetic analyses are highly sensitive to noise and because the slice spacing discrepancy is inconsistently present across the pedigreed sample, we decided that this level of known error was unacceptable.

To verify that the problem with slice spacing was in the DICOM files and not in the software 3D Slicer, we loaded the CT scan of skull W584 into Amira-Avizo (Thermo Fisher Scientific, Waltham, MA, USA). Due to proprietary nature of this software, it was not possible to see the slice spacing employed, but the same linear distance estimated between these two landmarks was 173.73 mm, which is more similar to the distance obtained in 3D Slicer with the 0.58 mm slice spacing model (173.3 mm) than to the 0.60 mm model (178.2 mm). Given that both Amira-Avizo and 3D Slicer automatically load this skull scan with a 0.584 mm slice spacing, we concluded that the problem is rooted in the DICOM files and not in the software. However, this conclusion did not determine which slice spacing value is correct, thereby providing the most accurate reflection of the physical skull. Fortunately, the physical skulls are still available for study, which allowed us to solve this puzzle.

Comparison of linear measurement from virtual models and real skulls

Next, we compared a highly replicable standard linear measurement from the CT scans and from the original skeletal specimens. Two landmarks were selected to calculate this linear distance. The first was prosthion (PR), defined as the most anterior point on the lingual surface of maxillary central incisor septum. The second landmark was basion (BA), defined as the midline point on the anterior margin of the foramen magnum (Fig. 1).

Figure 1 Inferior view of the cranium of baboon W281.

The dashed white line indicates the linear distance used to compare measurements obtained from the original skulls (using calipers and the Microscribe digitizer) with those derived from CT-based models. Anatomical landmarks are labeled as follows: PR, Prosthion; BA, Basion; OP, Opisthion; and LD, Lambda.

Although the PR-BA distance does not capture the maximum length of the skull, which is typically calculated as the distance from PR to lambda (LD), BA was selected instead of LD due to the methods employed during necropsy. The neurocranium of most of these skulls was sectioned to extract the brain, and in many cases, the posterior part of the foramen magnum was damaged, obliterating the opisthion (OP) landmark. Although the sectioned portion was reattached during the CT scan process to best approximate the amount of bone lost during sectioning, the reattachment process is a likely source of measurement error. Therefore, the PR-BA distance is a more reliable measurement of cranial length.

We created three sub-samples from the CT scans, comprising a total of twenty skulls: ten were scanned with a slice spacing of 0.75 mm, and the other ten with a spacing of 0.60 mm. Ten individuals were selected from the scans that had a slice spacing of 0.75 mm and returned no warning message related to slice spacing issues when imported with 3D Slicer (W001–W487). The Group 1 specimen numbers were: W023, W031, W096, W188, W264, W281, W297, W343, W451 and W481. For Group 2, we used the same sample() function to randomly select five individuals from the CT scans that have a slice spacing of 0.60 mm and no slice spacing error message when loaded into 3D Slicer (W670–W985). The Group 2 specimen numbers were: W728, W781, W805, W914, and W955. Finally, for Group 3, we randomly selected five individuals between W488 to W669, the specimens for whom a slice spacing error was returned when loading into 3D Slicer, and for which we are unsure if the slice spacing is 0.60 mm (as per the DICOM file heading) or 0.58 mm (automatically calculated by 3D Slicer and Amira-Avizo). The Group 3 specimen numbers were: W489, W535, W614, W620, and W650. We randomly assigned individuals to their respective groups using the function sample() from the base package in R (R Core Team, 2018).

The PR-BA distance was iteratively measured 10 times for each specimen format, and therefore, 20 times for each individual in Group 1 (CT scan and physical skull) and 30 times for each individual in Groups 2 and 3. For individuals in Groups 2 and 3, the distance was measured 10 times from the CT scan with a 0.58 mm slice spacing, 10 more times from the CT scan with a 0.60 mm slice spacing, and 10 times from the physical skull. In Group 2, the manually introduced spacing is 0.58 mm, whereas it was 0.60 mm in Group 3. All CT data were collected by study author M.M.-M. All skull data were collected by coauthor K.L.K by using a caliper.

Additionally, we compared our CT-derived measurements to the PR-BA distances calculated from landmark data collected by Joganic et al. (2018) from the physical skulls using a microscribe MS digitizer (Revware Inc., Raleigh, NC, USA). The raw data with all the measurements can be downloaded from the Supplemental Information 2).

The study employed two statistical approaches to compare measurements. First, a paired t-test was applied to assess whether the medians of two independent samples (e.g., physical measurements vs. tomographic measurements) were statistically distinct. Second, we evaluated whether the single microscribe MS measurements reported by Joganic et al. (2018) fell within the 99% prediction interval for each sample group. Both analyses were performed using R functions: the t_test() function from the rstatix package (Kassambara, 2023), with and paired = TRUE for the first approach, and the predict() function of the stats package (R Core Team, 2018) to estimate prediction intervals for the second approach. The R script code used to create figures and run analyses can be downloaded from Supplemental Information 3.

Results

The comparison of the repeated PR-BA measurements for Group 1 are shown in Fig. 2. For seven individuals the distribution of repeated measurements taken from CT scans with a slice spacing of 0.75 mm overlap the interquartile range of the distribution of repeated measurements taken from the physical skulls. However, for one of the remaining three specimens (W096), the CT-derived measurements overlap only with the smallest of the caliper-derived measurements. Conversely, the CT-derived data for W451 were on average 1.5 mm larger than those measurements taken from the skull, and there is no overlap in the distribution of the two types of measurements. Finally, for W481, the largest CT measurement is almost identical to the smallest measurement made on the skull, with the means of the two data sources differing by almost 1 mm. On average, within this group of ten specimens, the mean of the CT-derived PR-BA distance differs from that of the caliper-derived measurements by 0.18 mm. For six of the specimens, the microscribe-derived measurements are larger than any of the repeated measurements taken from the CT scans or the physical skulls. For three specimens, the measurements obtained from all three sources overlap. For the remaining specimen (W481), the microscribe measurement falls in the upper range of values measured from the physical skull. All these differences represent less than 1% of the average measurement.

Figure 2 Boxplots illustrating BA–PR distances across models for crania in Group 1.

This group comprises 10 individuals randomly selected from specimens W001 to W487, all with a CT slice spacing of 0.75 mm. The CT scans for these individuals did not generate any warning messages related to slice spacing artifacts. For each individual, the BA–PR distance was measured ten times on the CT-derived model (CT_0.75), ten times on the corresponding physical skull (Original), and once using the Microscribe digitizer, as reported by Joganic et al. (2018). Notably, the range of BA–PR distances (Y-axis) spans only 1.5 mm between minimum and maximum values. The selected specimens were: W023, W031, W096, W188, W264, W281, W297, W343, W451, and W481.

The boxplots of the repeated measurements for each of the five specimens in Group 2 are shown in Fig. 3. For these specimens, the CT scan, physical skull, and microscribe data provided almost the exact same measurements for both W955 and W914. For W805, the CT and microscribe measurements are almost identical, but the physical skull yielded a measurement that is 1 mm smaller. For W728 and W781, the physical skull measurement is between 0.5 and 1 mm smaller than that derived from the CT scan, and approximately 1.5 mm smaller than the measurement calculated from the microscribe data. As a reminder, for this group, the automatic slice spacing set by both software packages and the slice spacing indicated in the DICOM header are both 0.6 mm. When we manually re-set the slice spacing to 0.58 mm, PR-BA is 3–4 mm smaller than the measurements taken using a slice spacing of 0.6 mm, with a microscribe, or from the physical skull. Thus, 0.6 mm is the appropriate slice spacing for individuals W670–W985.

Figure 3 Boxplots illustrating BA–PR distances across models for crania in Group 2.

This group includes five individuals randomly selected from specimens W670 to W985, all with a CT slice spacing of 0.60 mm. None of the CT scans for these individuals generated warning messages related to slice spacing. For each individual, the BA–PR distance was measured ten times on the CT-derived model using the default slice spacing of 0.60 mm (CT_0.60), ten times on a model with the manually adjusted spacing of 0.58 mm (CT_0.58), ten times on the corresponding physical skull (Original), and once using the Microscribe digitizer, as reported by Joganic et al. (2018). The range of BA–PR distances (Y-axis) across all models is approximately five mm. The selected specimens were: W728, W781, W805, W914, and W955.

The results for Group 3 are presented in Fig. 4. For these specimens, we see the reverse of what was observed for Group 2. Here, the measurements from the CT scans with slice spacing manually set to 0.6 mm were more than 3 to 5 mm greater than the values obtained from the CT scans with 0.58 slice spacing automatically set, the physical skulls, and the microscribe.

Figure 4 Boxplots illustrating BA–PR distances across models for crania in Group 3.

This group comprises five individuals randomly selected from specimens W488 to W669, all of which had a reported CT slice spacing of 0.60 mm. However, loading these scans into 3D Slicer triggered a slice spacing error, creating uncertainty as to whether the true slice spacing is 0.60 mm (as indicated by the manually entered DICOM metadata) or 0.58 mm (as automatically inferred by 3D Slicer and Amira-Avizo). For each individual, the BA–PR distance was measured ten times on the CT-derived model using the automatically loaded spacing of 0.58 mm (CT_0.58), ten times using the manually corrected spacing of 0.60 mm (CT_0.60), ten times on the corresponding physical skull (Original), and once using the Microscribe digitizer, as reported by Joganic et al. (2018). The range of BA–PR distances (Y-axis) across all models is approximately five mm. The selected specimens were: W489, W535, W614, W620, and W650.

The results of the paired t-tests between all the samples in the three groups are shown in Table 1. In Group 1, the comparison of PR-BA distances between the original measurements and the CT_0.75 mm ones were not statistically different (p > 0.05), with the exception of W451 (p < 0.0001). In Group 3, comparisons between the original measurements and the CT_0.58 mm scans across all skulls revealed no statistically significant differences (p > 0.05). In contrast, comparisons between the original measurements and the CT_0.60 mm scans, as well as between the CT_0.58 mm and CT_0.60 mm scans, showed highly statistically significant differences (p < 0.0001). In Group 2, the situation is the opposite as in Group 3. When distances between CT_0.60 mm and the original measurements are compared, they are not statistically different (p > 0.05), with the exception of skull W805 (p < 0.0001). The remaining pairwise comparisons were significantly different (p < 0.0001).

Table 1 Paired t-tests in each baboon skull between all the models in Group 1, Group 2, and Group 3.

Comparisons were conducted among all available models within each Group: physical skull (Original) and the CT models with 0.58, 0.60 and 0.75 mm (CT_0.58, CT_0.60 and CT_0.75, respectively). The two models that were compared and their associated sample sizes are provided in the columns labeled Group 1, Group 2, n1, and n2, respectively. The output of the paired t-test is provided in the statistic column, along with the degree of freedom (df), p- value (p), the adjusted p-value (p.adj), and the significance of the adjusted p-value (p.adj.signif). The interpretation of the significance is: ns, not significant (p > 0.05); * (p ≤ 0.05); ** (p ≤ 0.01); *** (p ≤ 0.001); **** (p ≤ 0.0001).

Group	WUSTL	Group 1	Group 2	n1	n2	Statistic	df	p	p.adj	p.adj.signif	
Group 1	W23	CT_0.75	Original	10	10	2.90	9.00	0.02	0.72	ns	
W31	CT_0.75	Original	10	10	2.26	9.00	0.05	1.00	ns	
W96	CT_0.75	Original	10	10	−3.70	9.00	0.01	0.20	ns	
W188	CT_0.75	Original	10	10	2.15	9.00	0.06	1.00	ns	
W264	CT_0.75	Original	10	10	1.15	9.00	0.28	1.00	ns	
W281	CT_0.75	Original	10	10	1.46	9.00	0.18	1.00	ns	
W297	CT_0.75	Original	10	10	0.44	9.00	0.67	1.00	ns	
W343	CT_0.75	Original	10	10	1.84	9.00	0.10	1.00	ns	
W451	CT_0.75	Original	10	10	16.31	9.00	0.00	0.00	****	
W481	CT_0.75	Original	10	10	−3.71	9.00	0.01	0.20	ns	
Group 2	W728	CT_0.58	CT_0.60	10	10	−37.64	9.00	0.00	0.00	****	
CT_0.58	Original	10	10	−20.89	9.00	0.00	0.00	****	
CT_0.60	Original	10	10	4.49	9.00	0.00	0.08	ns	
W781	CT_0.58	CT_0.60	10	10	−55.32	9.00	0.00	0.00	****	
CT_0.58	Original	10	10	−29.20	9.00	0.00	0.00	****	
CT_0.60	Original	10	10	4.47	9.00	0.00	0.08	ns	
W805	CT_0.58	CT_0.60	10	10	−64.20	9.00	0.00	0.00	****	
CT_0.58	Original	10	10	−21.72	9.00	0.00	0.00	****	
CT_0.60	Original	10	10	12.59	9.00	0.00	0.00	****	
W914	CT_0.58	CT_0.60	10	10	−67.35	9.00	0.00	0.00	****	
CT_0.58	Original	10	10	−29.24	9.00	0.00	0.00	****	
CT_0.60	Original	10	10	1.86	9.00	0.10	1.00	ns	
W955	CT_0.58	CT_0.60	10	10	−57.24	9.00	0.00	0.00	****	
CT_0.58	Original	10	10	−23.43	9.00	0.00	0.00	****	
CT_0.60	Original	10	10	−0.84	9.00	0.43	1.00	ns	
Group 3	W489	CT_0.58	CT_0.60	10	10	−73.29	9.00	0.00	0.00	****	
CT_0.58	Original	10	10	−3.17	9.00	0.01	0.44	ns	
CT_0.60	Original	10	10	19.76	9.00	0.00	0.00	****	
W535	CT_0.58	CT_0.60	10	10	−57.41	9.00	0.00	0.00	****	
CT_0.58	Original	10	10	−0.42	9.00	0.68	1.00	ns	
CT_0.60	Original	10	10	25.26	9.00	0.00	0.00	****	
W614	CT_0.58	CT_0.60	10	10	−83.32	9.00	0.00	0.00	****	
CT_0.58	Original	10	10	1.01	9.00	0.34	1.00	ns	
CT_0.60	Original	10	10	14.81	9.00	0.00	0.00	****	
W620	CT_0.58	CT_0.60	10	10	−80.90	9.00	0.00	0.00	****	
CT_0.58	Original	10	10	1.44	9.00	0.18	1.00	ns	
CT_0.60	Original	10	10	28.68	9.00	0.00	0.00	****	
W650	CT_0.58	CT_0.60	10	10	−42.83	9.00	0.00	0.00	****	
CT_0.58	Original	10	10	−0.03	9.00	0.98	1.00	ns	
CT_0.60	Original	10	10	18.37	9.00	0.00	0.00	****	

The microscribe-derived measurements taken by Joganic et al. (2018) from the physical skulls offer an opportunity to further test the fit between caliper-derived measurements and the various CT model-derived measurements. We analyzed these microscribe measurements to see if they correspond with the prediction intervals of the other measurement techniques (confidence = 0.99) for any of the four sample groups (original, CT_0.75 mm, CT_0.60 mm, and CT_0.58 mm) (Tables 2 and 3). Based on statistical tests shown in Table 1, we expect that:

Table 2 Prediction intervals (upper and lower limits, confidence = 0.99) for Group 1 represented in the boxplots of Fig. 2.

The microscribe measurement for each individual is within the range of the prediction intervals from the Sample (TRUE) or (FALSE). Microscribe, measurement derived from the microscribe landmark data; Sample, the different data- collection methods; CT_0.75, measurements taken from the model of the skull created with a slice spacing of 0.75 mm; Original, measurements taken from the physical skull using calipers. All measurements in mm.

				Prediction interval (0.99)		
Group	WUSTL	Microscribe	Sample	Lower	Upper	Included	
Group 1	W23	122.82	CT_0.75	121.94	122.96	TRUE	
Original	121.10	123.28	TRUE	
W31	113.21	CT_0.75	112.48	113.88	TRUE	
Original	112.14	113.69	TRUE	
W96	123.72	CT_0.75	121.73	122.53	FALSE	
Original	121.02	124.44	TRUE	
W188	126.28	CT_0.75	125.47	126.45	TRUE	
Original	124.37	127.04	TRUE	
W264	116.67	CT_0.75	114.90	117.10	TRUE	
Original	114.84	116.77	TRUE	
W281	122.48	CT_0.75	121.34	124.06	TRUE	
Original	121.10	123.83	TRUE	
W297	117.16	CT_0.75	116.31	117.61	TRUE	
Original	115.19	118.56	TRUE	
W343	132.19	CT_0.75	130.74	131.76	FALSE	
Original	129.66	132.30	TRUE	
W451	147.20	CT_0.75	145.64	147.22	TRUE	
Original	143.50	146.34	FALSE	
W481	124.82	CT_0.75	122.75	124.33	FALSE	
Original	122.86	125.57	TRUE	

Table 3 Prediction intervals (upper and lower limits, confidence = 0.99) for Groups 2 and 3 represented in the boxplots of Figs. 3–4.

The microscribe measurement for each individual is within the range of the prediction intervals from the Sample (TRUE) or (FALSE). Microscribe, measurement derived from the microscribe landmark data; Sample, the different data-collection methods; CT_0.60, measurements taken from the model of the skull created with a slice spacing of 0.60 mm; CT_0.58, measurements taken from the model of the skull created with a slice spacing of 0.58 mm; Original, measurements taken from the physical skull using calipers. All measurements in mm.

				Prediction interval (0.99)		
Group	WUSTL	Microscribe	Sample	Lower	Upper	Included	
Group 2	W728	145.78	CT_0.60	143.63	146.45	TRUE	
Original	142.77	145.61	FALSE	
CT_0.58	139.83	142.39	FALSE	
W781	118.88	CT_0.60	117.55	118.77	FALSE	
Original	116.42	118.91	TRUE	
CT_0.58	114.39	115.13	FALSE	
W805	121.29	CT_0.60	120.75	121.89	TRUE	
Original	119.31	121.11	FALSE	
CT_0.58	117.27	118.35	FALSE	
W914	117.29	CT_0.60	117.03	117.83	TRUE	
Original	116.19	118.25	TRUE	
CT_0.58	113.71	114.43	FALSE	
W955	127.9	CT_0.60	126.98	128.32	TRUE	
Original	126.11	129.51	TRUE	
CT_0.58	123.6	124.42	FALSE	
Group 3	W489	122.07	CT_0.60	124.35	125.57	FALSE	
Original	120.64	123.5	TRUE	
CT_0.58	120.78	122.3	TRUE	
W535	118.92	CT_0.60	120.89	122.73	FALSE	
Original	117.23	119.62	TRUE	
CT_0.58	117.54	119.18	TRUE	
W614	118.61	CT_0.60	121.34	122.4	FALSE	
Original	115.8	120.69	TRUE	
CT_0.58	118.21	118.75	TRUE	
W620	152.67	CT_0.60	155.43	157.31	FALSE	
Original	150.21	153.14	TRUE	
CT_0.58	151.22	152.56	FALSE	
W650	141.33	CT_0.60	144	146.4	FALSE	
Original	138.68	143.71	TRUE	
CT_0.58	140.6	141.78	TRUE	

The microscribe-derived measurements are within the prediction limits in all the caliper-derived measurements taken from the physical skulls in Groups 1, 2 and 3. We found that the microscribe-derived measurements are within the prediction interval of the caliper-derived measurement for 17 out of 20 skulls (85%, Tables 2 and 3). The exceptions are W451 (Group 1, Table 2) and W728 and W805 (both in Group 2, Table 3). For these three individuals, the microscribe-derived measurements are within less than one mm of the upper limits of the caliper-derived measurements (0.86 mm, 0.17 mm and 0.18 mm, respectively). In terms of anatomical variation, this represents less than 1% of the overall measurement. Therefore, even though the difference is statistically significant, the distinction between the microscribe- and caliper-derived does not represent a significant amount of measurement error from an anatomical sciences perspective.

The microscribe-derived measurements are within the range of values calculated from the CT-0.75 mm models of skulls from Group 1 (i.e., the correct slice spacing). We found that the microscribe-derived measurements are within the limits of the CT-0.75 mm sample for 8 out of 10 individuals, with the exception of W096 and W343 (Table 2). For these two individuals, the microscribe-derived values are 1.19 mm and 0.43 mm above the upper limits of the CT_0.75 mm model-derived values, respectively. As noted in the previous paragraph, although statistically significant, this does not represent measurement error greater than 1%.

The microscribe-derived measurements are within the range of values calculated from the CT_0.60 mm models of skulls from Group 2 (i.e., the correct slice spacing). As expected, for 4 out 5 skulls, the microscribe-derived measurement is within the range of values obtained from the CT_0.60 mm model (Table 3). The one exception is W781, for which the microscribe-derived value is 0.11 mm above the upper end of the CT_0.60 mm model-derived values (less than 1% measurement error).

The microscribe-derived measurements are within the range of values calculated from the CT_0.58 mm in Group 3 (i.e., the correct slice spacing). For four of five individuals, the microscribe-derived measurement is within the range of values from the CT scan models set to a slice spacing of 0.58 mm (Table 3). The one exception is W620, whose microscribe measurement is also 0.11 mm above the upper limit (less than 1% measurement error).

The microscibe-derived measurements are below the measurements derived from the CT_0.60mm model for individuals in Group 3 (as the slice spacing should be 0.58 mm). For all five individuals, the microscribe measurements are 1.97 mm to 2.76 mm below the range of values obtained from the CT_0.60 mm models (Table 3).

The microscribe-derived measurements are above the measurements derived from the CT_0.58mm model for individuals in Group 2. As expected, all five skulls are 2.86 mm to 3.75 mm above the prediction upper limits of CT_0.58 mm samples (Table 3).

Overall, our results indicate that in 43 out of 50 comparisons (86%), the microscribe-derived data fit either within or very close to the ranges of both the caliper-derived measurements and those taken from the CT models with the correct slice spacing. As the microscribe data are either significantly larger or smaller than the measurements taken from the CT models with incorrect slice spacing, our designations of the correct identity of the slice spacing values for the three groups are supported.

In conclusion, our results show that the automatically calculated slice spacings in 3D Slicer faithfully represents the physical skulls (0.75 mm for Group 1, 0.60 mm for Group 2, and 0.58 mm for Group 3).

Discussion

Here, we described the investigation of a mismatch between manually reported and automatically detected slice spacing for a subset of CT scans acquired for 985 SNPRC baboon skulls. We walked through our process for identifying the source of the mismatch (the metadata reported with the associated DICOM files on the data aggregation site), and our process for determining which CT slice spacing value is the best match for the physical skull, using both caliper- and microscribe-derived measurements for reference. We found that the automatic detection of slice spacing in both 3D Slicer and Amira-Avizo creates a 3D model that best reflects the physical skull, although this value is hidden in the proprietary Amira-Avizo and made obvious in 3D Slicer.

The use of virtual paleontology is significantly increasing (Cunningham et al., 2014). Although there are some studies comparing physical and virtual measurements (Tolentino et al., 2018), different virtual models of the same fossils (Díez Díaz et al., 2021), or the comparison of different parameters of the same virtual model to find the optimal combination (Pérez-Ramos & Figueirido, 2020), most scientists with access to the virtual models do not have access to the original specimen. Therefore, DICOM files, surface meshes, or other 3D data must be assumed to be a faithful representation of the original given that this assumption cannot easily be tested.

The great benefits of these virtual models are undeniable. For example, scans can help to limit the handling of valuable and irreplaceable specimens (Gilissen, 2009). They also make access to specimens more equitable to those with access to the internet and adequate digital storage. However, scientists must be mindful that they are dealing with a digital model of the original specimen that is the result of many procedural steps (Noumeir & Pambrun, 2012; Li et al., 2016). Each parameter value that drives the creation of the virtual model has the potential to contribute to the creation of an unrealistic or unfaithful model, compromising its scientific value.

We were unable to contact the technicians that scanned the SNPRC baboon CT scans of specimens W488 to W669, so we suspect that in the process of scanning or of converting the raw file into DICOM files, some parameters may have been unintentionally modified, leading to the slice spacing issues. 3D Slicer brought this issue to our attention with a warning message. Although Amira-Avizo automatically detected the most appropriate slice spacing, this adjustment was invisible to the user, which could lead to unforeseen complications for investigators less familiar with scanning protocols.

It is critical that such analytical decisions made by any software are made transparent and any issues encountered during data import are communicated clearly. For instance, while Amira-Avizo appeared to handle the data without errors and loaded the data with a seemingly correct spacing value, it did so without notifying the user of any assumptions or modifications. In contrast, 3D Slicer identified a spacing inconsistency and halted the import process until the user explicitly confirmed how to proceed. Moreover, 3D Slicer clearly documented the adjustments it applied (i.e., regularizing slice spacing), thereby allowing the user to verify the changes. While automatic corrections can be helpful, undetected or incorrect assumptions—such as those potentially made by Amira-Avizo—pose a risk to data integrity. As done by 3D Slicer, any inconsistency in the data should be communicated to the user to ensure transparency and reproducibility.

In addition to determining the proper slice spacing value, our study also raises two other interesting points relevant to the use of 3D models in anatomical research. First, as is immediately evident in Figs. 2 and 3, every method of measurement collection includes error. Sometimes these different measurement collection approaches return almost identical values, but this was only the case for 6 of the 20 specimens (30%) included in our study. For the most part, microscribe data tend to return relative increases in the measurement value, whereas CT-derived measurements and caliper-derived measurements do not consistently yield relatively larger or smaller values compared to each other. Just over 50% of the individuals in our study reported smaller caliper-derived distances compared to the CT-derived distance. When looking at these results, we only measured one cranial dimension, so the box plots are reporting the variation in repeated measurements. This observation echoes the caution that has been voiced previously, in both industry and medical cases (Lascala, Panella & Marques, 2004; Lund, Gröndahl & Gröndahl, 2009; Carmignato, 2012). While these discrepancies could be worrisome, the reality is that the measurement differences are well below the level of measurement error that most anatomists consider acceptable (<3%) (Stull et al., 2014).

A second observation that is relevant to our results pertains to the confidence with which we imbue CT scan data. Ford & Decker (2016) ran a study of 20 human crania CT scanned with different slice thicknesses, finding that placing landmarks on models created with two mm or greater slice spacing returns questionable results. Fortunately, the SNPRC baboon skulls were all scanned well below this threshold, with slice spacing of 0.58 mm, 0.6 mm, or 0.75 mm. The key, as we discovered, lies in accurately determining which slice spacing value corresponds to each specimen. From our investigation, we conclude that:

• Individuals W001 to W487 were scanned with a slice spacing of 0.75 mm.

• Individuals W488 to W669 were scanned with a slice spacing of 0.58 mm.

• Individuals W670 to W985 were scanned with a slice spacing of 0.60 mm.

It cannot be overstated that the simplicity of this discovery greatly masks the amount of time required to resolve it. More than 250-person hours of effort, spread over 8 months were dedicated to resolving this issue. First, we had to trouble-shoot the import error, ultimately realizing that the slice spacing parameter was the cause. We then had to identify which specimens returned the import error, and therefore had a slice spacing mismatch. This turned into an investigation to ascertain which slice spacing value was correct. This effort ultimately reduced a 3–5% measurement bias in nearly 20% of the sample, primarily affecting measurements along the anterior-posterior axis.

Conclusions

While digital data aggregators such as MorphoSource provide invaluable access to morphological datasets, some level of quality control should be implemented to ensure the reliability of the shared data. Additionally, direct access to physical specimens remains essential; many analytical procedures and validations—such as those conducted in this study—would not have been possible without firsthand examination of the physical skulls.

Resolving this puzzle cost our research group hundreds of hours of researcher time. However, we are now aware of a source of measurement error that could have undermined our genetic analyses. We have also identified the correct slice spacing parameter values that can be shared with MorphoSource, so that these scans can be included in semi-automated data collection protocols.

We share this cautionary tale to strongly encourage investigators to take the time to ensure that they are familiar with the nuances of the scans from which they are collecting data (as we imagine the temptation to overlook the loading error would be strong). Additionally, we hope that our experience can be cited as justification to funding agencies when asking for the financial support needed to carefully process digital images to avoid errors, both on the data-collection side as well as the resource-sharing side of the process.

Supplemental Information

Supplemental Information 1 Supplemental Figures

Supplemental Information 2 Raw data with the PR-BA distances

Supplemental Information 3 R script to obtain Figures 2-4 and Tables 1-3

We thank the editor and the referees for their thoughtful and constructive critiques. We also thank the organization/staff managing the MorphoSource repository for keeping these (and other) relevant virtual specimens. We also thank to Marina Martínez de Pinillos for assistance with Amira/Avizo. The views and opinions expressed in this article are those of the author(s) alone and do not necessarily reflect the official policy or position of the United States government or any of its agencies, the European Union, or the European Research Council. Neither the European Union nor the granting authority can be held responsible for them. The information presented is based on the author(s)’ research and analysis and is intended for educational and informational purposes only, with the author(s) assuming full responsibility for the content and any conclusions drawn.

Additional Information and Declarations

Competing Interests

Author Contributions

Data Availability

The authors declare there are no competing interests.

Mario Modesto-Mata conceived and designed the experiments, performed the experiments, analyzed the data, prepared figures and/or tables, authored or reviewed drafts of the article, and approved the final draft.

Arthur Thiebaut performed the experiments, analyzed the data, authored or reviewed drafts of the article, and approved the final draft.

Kristin L. Krueger performed the experiments, analyzed the data, authored or reviewed drafts of the article, and approved the final draft.

A. Murat Maga analyzed the data, authored or reviewed drafts of the article, and approved the final draft.

Jessica L. Joganic performed the experiments, analyzed the data, authored or reviewed drafts of the article, and approved the final draft.

Timothy M. Ryan analyzed the data, authored or reviewed drafts of the article, and approved the final draft.

Joan T. Richtsmeier analyzed the data, authored or reviewed drafts of the article, and approved the final draft.

James M. Cheverud analyzed the data, authored or reviewed drafts of the article, and approved the final draft.

Leslea J. Hlusko conceived and designed the experiments, performed the experiments, analyzed the data, prepared figures and/or tables, authored or reviewed drafts of the article, and approved the final draft.

The following information was supplied regarding data availability:

The code and raw data are available in the Supplemental Files.

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
