# Peer review of "Easier said than done: unexpected hurdles to preparing ∼1,000 cranial CT scans for data collection from an online digital repository"

_PeerJ, doi:10.7717/peerj.20172_

## Round 0.1 · original submission · Major Revisions

·

Basic reporting

The basic reporting meets the criteria for PeerJ. The English is unambiguous, the references are sufficient, context is provided, the structure is sound, and the document is self-contained. However, I do have some minor comments that should improve the flow and presentation, as follows:

1. Line 70: Provide a citation for iDigBio.

2. Line 132: Replace "in order to be extra cautious when" with "for".

3. Line 133: The mention of best practices is confusing. It is unclear whether the best practices cover details such as consistent slice spacing. From looking at the Davies reference, the best practices are phrased as higher-level principles. My advice is to replace "by following the best practice set (Davies et al., 2017)" with "as part of following the best practices for publishing verified 3D digital data (Davies et al., 2017)."

4. Line 203, 216, 233, 236 (and possibly elsewhere): Replace "in order to" with "to". "in order to" is a low information content phrase. Using one word in place of three will make the writing tighter.

5. Throughout the paper, mention is made of Groups 1, 2, and 3. I suggest replacing the labels 1, 2, and 3 with something mnemonic, so readers don't have to remember which group is which. For instance, Groups 1, 2, and 3 could be relabelled as SS75_noWarn, SS60_noWarn, and SS?_Warn, respectively. A different naming convention would be fine, but selecting names that remind the reader will reduce their cognitive burden, letting them focus more on the paper itself.

6. Line 279: The meaning of MM-M and KLK wasn't immediately obvious. I suggest adding the phrase "study author". For instance, saying: "All CT data were collected by study author MM-M."

7. Lines 369 and 373: The use of the future tense is confusing. I suggest using the present tense. For instance, "The microscribe-derived measurements are below ...".

8. Line 387: Rather than Discussion, a more standard title for the last section of a paper would be Conclusions, or Concluding Remarks.

9. Lines 424 to 426: The text repeats the advantages of 3D Slicer's approach, which makes the paragraph feel repetitive. I suggest changing the last two sentences to one sentence: As done by 3D Slicer, any inconsistency in the data should be communicated to the user to ensure transparency and reproducibility.

10. Lines 436 to 437: Starting a clause with "it" can be a weak way to structure a sentence. I suggest deleting "it is important to remember that".

11. Line 439. I suggest changing "either in" to "in both".

12. Lines 458 to 460: The last sentence of this paragraph is on a different topic than the rest of the paper. The inclusion of this sentence hurts the cohesion of the original paragraph. I suggest deleting the last sentence. (The one starting with "Fortunately the original skulls ...".) Deleting this sentence does not remove information from the paper, since the point of this sentence is adequately covered in the paragraph from lines 465 to 469.

13. Figure 2, 3, and 4 captions: The spacing in the text of the captions is odd. For instance, "were" is spaced as "we re". (Similar spacing issues occur throughout the captions.) The phrase "Ten times was measured" is awkward. Also, the second-to-last sentence is awkward. The awkward sentence is the one that starts with "Realize that the difference ...". Who is this imperative directed toward?

14. Figure 2, plot labels: The labels for W23, W31, and W96 should be changed to W023, W031, and W096, respectively. This will bring these labels in line with how the data sets are labelled throughout the paper.

15. Tables 2 and 3: The underline for FALSE makes the tables busier than necessary. Using bold font for FALSE is enough to make it stand out.

Experimental design

The paper does a great job on the experimental design. The primary question (the appropriate slice spacing) is defined, and a rigorous approach is used to answer the question. The method description and supplementary data are adequate for someone else to replicate the work. The choices made in the method are justified. For instance, the choice of using the PR-BA distance is nicely explained in lines 252-258.

Validity of the findings

If impact and novelty were important review criteria, I would ask for the authors to present a general methodology that others can use to assess their 3D digital data. I might also criticize that the steps in the experimental design as not particularly novel. Fortunately, the journal recognizes the need for research of the sort presented in this paper. This paper is a great example of finding a problem with data and systematically addressing and fixing the problem, regardless of how long it takes to do it right. Other researchers will be able to follow this example. Moreover, if authors get academic credit for doing things the right way, it will be easier to foster a strong scientific community.

The conclusions are well-stated, but I do have two questions/requests:

1. The authors hypothesize about the sources of the inconsistencies in some of the data sets. For instance, in Lines 182 and 183, it is suggested that the different reports on slice spacing and the specific scanner are typographic mistakes by Atkinson. In lines 412 and 413, it is suggested that parameters may have been unintentionally modified. Have the authors made any effort to contact the original study authors to ask about the possibility of errors or other explanations? If it is not feasible to ask the original authors something to that effect should be stated in the paper.

2. On lines 50 and 454, mention is made of 250 person-hours of effort for resolving the discrepancy. The case for the validity of this result would be stronger if more detail was provided. What tasks went into the 250 hours? How long did each task take?

Additional comments

The authors of this paper deserve credit for treating their data with integrity. They spent considerable time investigating the "inconsistent slice spacing" error from 3D Slicer. With time and resource pressures, they could have just used 3D Slicer's automatic correction and glossed over the details in future publications. This wouldn't have been the right thing to do, but there is a good chance nobody would notice. Integrity is what you do when nobody is watching.

Should the authors make an explicit statement that the purpose of the paper is not for future researchers to justify the use of automatic detection of slice spacing? The purpose of the paper (as I read it) is for future researchers to thoroughly understand their digital datasets and question every potential inconsistency or error.

I like that the convention adopted for defining "slice spacing" is carefully explained in lines 164-169. This is very helpful for readers of the paper since they may usually employ a different assumption for the meaning of this phrase.

Reviewer 2 ·

Basic reporting

The structure of the manuscript is solid and adheres to PeerJ’s acceptable format; however, specific aspects of the manuscript could be improved.

Introduction and Background: The manuscript provides a thorough background of both the importance of digital data collection and curation, as well as the information regarding the specific dataset examined in this manuscript.

Language: The language used in this manuscript on occasion is casual rather than professional, for example:
Line 127 - “We share our journey in this manuscript for three reasons.”
Line 447-448 – “The trick, as we discovered, is making sure to know which slice spacing value applies to which specimen”
Table 2 legend – “The comparisons were done between all the available models.”

Multiple sentences in the materials and methods section use the present tense rather than the past tense to describe methodology. Examples include:
Line 192 - “involve”, Line 194 - “results”, Line 195 - “includes”, Line 196 - “generates”, Line 197 - “is”, Line 198 - “depends”, Line - 199 “visualize”, Line 200 - “realize” and Line 201 - “is”.
This is not an exhaustive list, and many more examples appear within the text that must be rectified.
The sentence on lines 328-331 is not grammatically correct.
The words “they all” on line 331 should be removed.
Overall, the grammar and spelling should be examined over the whole article.

Figures and Tables: The figures are of high quality, well-labelled, and correctly cited in the text. Figure 3 contains the additional text “Skull model” below the boxplot, which does not appear on other figures; this should be removed.

Multiple figure legends have grammatical and spelling errors
The Figure 2 legend contains the sentences:
“Ten times was measured the BA-PR distance in the CT model (CT_0.75), ten times in the physical skull (Original), and once with the micr os cribe, which comes from the Joganic et al. (2018) paper.”
“Realize that the difference between the minimum and maximum values of the BA-PR distance (Y axis) is only about 1.5 mm.”

Present tense is used in the figure legend as well
“The selected individuals are: W023, W031, W096, W188, W264, W281, W297, W343, W451 and W481.”
Supplemental Information Figure 7 describes part C, which appears to be missing.

Raw Data: The raw data and code have been included within the submission

Experimental design

Scope: The manuscript examines an aspect of using available datasets that has not been examined in the literature before. It represents original research that falls within the scope of the journal.

Methodology: The methodology of the manuscript is largely well described, however could require more details in some areas. Specifically, in regards to the orientation of the CT scans and the choice of using the prosthion to basion measurement. Although the slice spacing/thickness of the dataset has been well described, it is worth mentioning what the raw slice orientation is, as this affects which plane is affected by the slice spacing. The choice of a lengthwise measurement, i.e., prosthion to basion, is therefore justified compared to a measurement related to skull width or height. Which axes slice spacing affects if changed, should be specified.

Although it was mentioned that slice spacing and thickness will collectively be referred to as slice spacing, as they are the same in this dataset, it may offer more clarity to specify that 3D slicer automatically changed both slice spacing and slice thickness to 0.58 mm (if this was the case).

Validity of the findings

It was reported in the manuscript that values were compared via a pairwise Welch’s t-test. A Welch’s t-test assumes independence, so it cannot be used pairwise. The code provided in the supplemental information does not appear to have a paired set to true.

For the function t.test, the argument for paired is not set and is FALSE by default.

Running the code provided the same results as reported in the paper, and as such, there are no paired t-tests. Considering that the measurements are of the same skulls, a paired t-test should be used. The analysis should be rerun.

Lines 295-297 refer to an overlap of the CT scan repeated measurements overlapping with the “first confidence interval” of the repeated measurements from the original skulls within Figure 2.
Figure 2 contains box plots that do not appear to have confidence intervals. The code used to create the figure only creates boxplots, which contain the first and third quartiles rather than a confidence interval.
Line 319: The sentence “Therefore, we conclude that 0.6 mm is the appropriate slice spacing value for individuals between W670 to W98.” is a conclusion from the results, and thus belongs in the discussion.

Percent differences in measurement of 3-5% are mentioned in the abstract; however is not mentioned within the results or discussion, only metric lengths are. More emphasis on following metadata reported in the dataset would have led to erroneous measurements should be given.

Additional comments

Strengths:
The manuscript explores an aspect of dataset usage that has not been previously explored.
The authors have provided good background and context for the topic.
The authors have done an exemplary job of determining the correct slice spacing by tracking down and measuring the original skulls, allowing the dataset to be used accurately and without fear or error in the future.

Weaknesses:
Minor errors of language are present throughout and should be fixed.
The statistical tests should be rerun and reported correctly based on the data.

---

## Round 0.2 · Minor Revisions

Thank you for revising your manuscript. The revision makes the text easier to follow and of even broader relevance. I feel it is crucial to publish your study but there are still some important points which need to be addressed before publication:

1) Repository description/attribution: Given you are using data from MorphoSource, I feel it would be appropriate to cite a reference describing the repository as well as the website where this it can be found. In addition, although not mandatory, I feel it would also be appropriate to thank the repository staff/organization for maintaining it as your study would not have been possible without original authors depositing and staff/organization maintaining MorphoSource.

2) Description of performed statistical analysis: there are discrepancies between the description of the test in the methods and the analyses actually performed. The ones performed seem correct but differ from information provided in methods (see reviewer 2 who explains the necessary changes in greater detail). I would also like some further explanation as to why you deem 10 (5 + 5) specimens in the error group are sufficient to reveal the true slice spacing.

3) Formatting and typographical issues: there are some minor formatting and typographical issues. Please find them in the annotated pdf.

Please make sure to address these and all others points raised including those in annotated pdfs.

I look forward to receiving the revised manuscript.

·

Basic reporting

The authors have addressed my original concerns, with one exception (the labelling of the Groups). The one exception is not a concern for me. It can be considered a difference in style preference.

Through their edits, I did not see the addition of any new basic reporting issues.

Experimental design

The authors have addressed the concerns of the reviewers.

Validity of the findings

The authors have addressed the concerns of the reviewers.

Additional comments

The authors have addressed the concerns of the reviewers.

Reviewer 2 ·

Basic reporting

The authors have sufficiently changed the language and added sentences to address all comments.

Experimental design

Examination of the R code reveals the correct t-test has been conducted, a paired t-test. Although the code has been set to unequal variance, this setting is overridden by the paired = true argument, so the results are representative of a paired t-test. As such, the description of the t-test in the text describing should be changed to say a paired t-test was conducted, removing the Welch element (A paired Welch t-test does not exist). Additionally, the description describes the test as 'whether the medians of two independent samples (e.g., physical measurements vs. tomographic measurements) were statistically distinct". The test does not do this. The physical and tomographical measurement are dependent as they are a repeated measures measurement using two different methods. Secondly, the means or medians are not compared, rather the statistical significance of the mean difference is examined. Whilst the methods have been applied correctly and the results are valid, the description of the statistical test must be changed. All other comments have been addressed.

Validity of the findings

The authors have addressed or successfully rebutted my comments.

---

## Round 0.3 · accepted · Accept

Thank you for addressing our suggestions. The revisions make the manuscript even easier to follow and of even broader relevance. I feel it is a great contribution to the field highlighting the constraints and solutions to think about when preparing and/or re-using large openly accessible CT-data sets. I look forward to see this published!